# Telomere Length and Mitochondrial DNA Copy Number Variations in Patients with Obesity: Effect of Diet-Induced Weight Loss—A Pilot Study

**DOI:** 10.3390/nu14204293

**Published:** 2022-10-14

**Authors:** Raffaella Cancello, Federica Rey, Stephana Carelli, Stefania Cattaldo, Jacopo Maria Fontana, Ilaria Goitre, Valentina Ponzo, Fabio Dario Merlo, Gianvincenzo Zuccotti, Simona Bertoli, Paolo Capodaglio, Simona Bo, Amelia Brunani

**Affiliations:** 1Obesity Unit and Laboratory of Nutrition and Obesity Research, Department of Endocrine and Metabolic Diseases, Istituto Auxologico Italiano, IRCCS, 20121 Milano, Italy; 2Pediatric Research Center “Romeo ed Enrica Invernizzi”, Department of Biomedical and Clinical Sciences, University of Milano, 20157 Milano, Italy; 3Laboratory of Clinical Neurobiology, Istituto Auxologico Italiano, IRCCS, Piancavallo, 28824 Verbania, Italy; 4Laboratory of Biomechanics, Rehabilitation and Ergonomics, Istituto Auxologico Italiano, IRCCS, Piancavallo, 28824 Verbania, Italy; 5Dietetic and Clinical Nutrition Unit, Città della Salute e della Scienza Hospital of Torino, 10126 Torino, Italy; 6Department of Pediatrics, Children’s Hospital “V. Buzzi”, University of Milan, 20154 Milano, Italy; 7International Center for the Assessment of Nutritional Status (ICANS), Department of Food, Environmental and Nutritional Sciences (DeFENS), University of Milan, Via Sandro Botticelli 21, 20133 Milano, Italy; 8Department of Surgical Sciences, Physical Medicine and Rehabilitation, University of Turin, 10126 Torino, Italy; 9Department of Medical Sciences, University of Turin, Corso Dogliotti 14, 10126 Torino, Italy

**Keywords:** telomeres, telomeres length, mitochondrial DNA copy number, obesity, weight loss, diet, nutrition, total antioxidant capacity

## Abstract

Background: Telomere length (TL) and mitochondrial DNA (mtDNA) copy number shifts are linked to metabolic abnormalities, and possible modifications by diet-induced weight loss are poorly explored. We investigated the variations before (T0) and after a 1-year (T12) lifestyle intervention (diet + physical activity) in a group of outpatients with obesity. Methods: Patients aged 25–70 years with BMI ≥ 30 kg/m^2^ were enrolled. Clinical and biochemical assessments (including a blood sample for TL, mtDNA copy number and total antioxidant capacity, and TAC determinations) were performed at T0 and T12. Results: The change in TL and the mtDNA copy number was heterogeneous and not significantly different at T12. Patients were then divided by baseline TL values into lower than median TL (L-TL) and higher than median TL (H-TL) groups. The two groups did not differ at baseline for anthropometric, clinical, and laboratory characteristics. At T12, the L-TL group when compared to H-TL showed TL elongation (respectively, +0.57 ± 1.23 vs. −2.15 ± 1.13 kbp, *p* = 0.04), higher mtDNA copy number (+111.5 ± 478.5 vs. −2314.8 ± 724.2, respectively, *p* < 0.001), greater weight loss (−8.1 ± 2.7 vs. −6.1 ± 4.6 Kg, respectively, *p* = 0.03), fat mass reduction (−1.42 ± 1.3 vs. −1.22 ± 1.5%, respectively, *p* = 0.04), and increased fat-free mass (+57.8 ± 6.5 vs. +54.9 ± 5.3%, respectively, *p* = 0.04) and TAC levels (+58.5 ± 18.6 vs. +36.4 ± 24.1 µM/L, respectively, *p* = 0.04). Conclusions: TL and the mtDNA copy number significantly increased in patients with obesity and with lower baseline TL values after a 1-year lifestyle intervention. Larger longitudinal studies are needed to confirm the results of this pilot study.

## 1. Introduction

Telomeres are specialized structures at the ends of linear chromosomes that play a relevant role in regulating cellular replicative capacity. The scientific literature has reported that accelerated telomere attrition and short telomere length (TL) mediate many age-related pathological conditions, such as impaired glucose tolerance, insulin resistance, obesity, type 2 diabetes mellitus (T2DM), and the metabolic syndrome [1,2,3,4,5,6]. The involved molecular mechanisms are still under investigation.

The TL varies between individuals of the same age and appears to be linked to a wide number of environmental factors (i.e., gender, ethnicity, stress, physical activity, obesity, smoking, alcohol consumption) [7]. The shortening of telomeres is somehow balanced by the activity of telomerase, expressed in highly replicating tissues, such as stem cells, germ cells, and activated lymphocytes [8]. Although telomeres are known to be genetically determined (heritability ranging from 44% to 80%), telomere shortening is inversely associated with oxidative stress [9] and chronic low-grade inflammatory state (especially the so called “inflamm-aging”) [10]. Obesity can lead to oxidative stress and increased systemic inflammation, which can both determine telomere shortening. Patients with obesity have shorter telomeres than lean individuals [11]. Excess adiposity increases the production of pro-inflammatory cytokines that sustain the chronic low-grade inflammatory state, thus favoring both dysmetabolic conditions and aging with earlier onset of age-related diseases [12,13].

The mtDNA, a circular multicopy genome DNA located inside the mitochondrion, is also involved in the regulation of the aging process, being susceptible to the structural damage caused by various factors, such as oxidative stress and inflammation [14,15]. A potential bidirectional relationship between TL and the mtDNA copy number modification was observed in a population-based study showing that mtDNA was inversely associated with BMI and positively with TL [16]. Furthermore, mtDNA was not associated with age, while TL was negatively associated with age even after adjusting for the mtDNA copy number cell content [16]. There is evidence that TL linearly decreases with weight gain, and people with obesity presented shorter TL than those with a stable normal weight [17].

Different types of weight-loss interventions produce different TL modifications [18]. Greater weight loss after bariatric surgery induced a greater TL in subjects with the shortest telomeres at baseline [19]. The consumption of foods rich in antioxidants (such as olive oil, fruits and vegetables, tea, wine, etc.), adherence to the Mediterranean diet (MD), smoking, physical activity, and stress can impact on TL too. Increased systemic oxidative stress has been associated with shorter TL in leukocytes [20], but an in vivo experiment provided controversial associations between oxidative stress and telomere attrition due to a limited number of studies [21]. Physically inactive adults showed shorter TL compared to active peers [22]. Furthermore, regular exercise was associated with a greater leukocyte mtDNA copy number in postmenopausal women [23]. Globally, the relationship between obesity and the TL/mtDNA copy number in adults has shown controversial results [11,24] and there are few data available on the effect of diet-induced weight loss.

The primary objective of the present pilot study was therefore to evaluate the effects of a lifestyle intervention (diet plus exercise) in a real-life situation, on TL and the mtDNA copy number in a cohort of adult outpatients with severe obesity. Secondary objectives were to assess the presence of possible correlations between changes in clinical and laboratory variables and the TL/mtDNA copy number modifications in the same cohort.

## 2. Materials and Methods

### 2.1. Subjects

We performed an observational pilot study. Outpatients aged 25–70 years, with BMI ≥ 30 kg/m^2^ were enrolled at the Obesity Unit of the “*Città della Salute e della Scienza*” Hospital of Torino after signing a written informed consent. Patients classified as EOS (Edmonton obesity staging system) stage ≥2, with secondary obesity, at risk of eating behavioral or psychiatric disorders (evaluated by the Hamilton Rating Scale for Depression [25], the Hamilton Anxiety Scale [26] and the Binge Eating Scale [27]), who were candidates for bariatric surgery were excluded.

### 2.2. Lifestyle Intervention

All patients received dietary recommendations in line with the principles of the Mediterranean diet (MD), with an average calorie intake of ~1500 ± 100 kcal/die (protein 15–20%; lipids 25–30%; carbohydrates 50–60%, 30 g fibers) and the indication for varying foods according to personal eating habits, taste, food preferences, cultural traditions, and work-life habits. Furthermore, each patient received indication for regular physical exercise: 20 min of walking/day, 10 min of resistance training at least 3 times/week. The participants were followed up every 3 months (at 3, 6, 9, and 12 months after entering the study) by a dietitian and a medical doctor. Subjects who withdrew from the study for any reason before 12 months, or those who were on medications for weight control or under weight management procedures other than those recommended (e.g., very low-calorie diets, highly unbalanced diets) were considered dropouts.

### 2.3. Measurements

At enrollment (T0) and after 12 months (study end, T12), all the participants underwent the following assessments: the Minnesota-Leisure-Time-Physical-Activity questionnaire [28], measurements of body weight (kg), height (cm), waist (cm), body composition by dual-energy X-ray absorptiometry (DEXA), arterial blood pressure, and blood circulating markers. A dietary interview was performed to calculate at baseline the mean daily caloric intake and monitored at follow-up. Physical activity level was calculated as the product of the duration and frequency of each activity (hours/week), weighted by an estimate of the metabolic equivalents (METs) of the activity and summed for all the activities performed [29].

Venous blood samples after an overnight fast were collected in heparinized vacutainer tubes. After centrifugation at 1500× *g* at 4 °C for 5 min, plasma was collected and transferred in Eppendorf tubes and then immediately frozen and stored at −80 °C until analysis. A dedicated tube was collected and immediately frozen for DNA extraction. At T0 and T12, fasting glucose, insulin, glycated hemoglobin (Hb1Ac), total and high-density lipoprotein cholesterol, triglycerides, and high-sensitivity hs-C-reactive protein (hs-CRP) were assessed using an automated analyzer (Roche Diagnostics, Mannheim, Germany). The homeostasis model assessment of insulin resistance index (HOMA-IR) was calculated with the formula: fasting plasma glucose (mmol/L) by fasting serum insulin (mU/L) divided by 22.5 [30].

Total antioxidant capacity (TAC) assay was employed to determine the overall antioxidant status by using a commercially available kit based on photometric technique (ImmunDiagnostik^®^, Bensheim, Germany). The sensitivity for TAC testing was determined by high-performance liquid chromatography (HPLC) using kits from Chromsystems (Chromsystems Instruments & Chemicals, Gräfelfing, Germany) and following manufacturer instructions. The sensitivity for TAC testing was 130 µmol/L, the intra- and inter-assay coefficient of variations (CVs) were <3.5%. Serum malonyldialdehyde (MDA) concentrations were determined as end products of lipid peroxidation by HPLC (using kits from Chromsystems Instruments & Chemicals, Gräfelfing, Germany) and following manufacturer instructions. The sensitivity was 0.01 µmol/L and the intra- and inter-assay CVs were <8%.

### 2.4. Telomere Length and Mitochondrial DNA Copy Number Determination

Genomic DNA was extracted from blood PBMC using a commercial kit (Puregene Blood Core Kit B, Qiagen, Minneapolis, MN, USA) and following manufacturer instructions. Quantification and purity of DNA was performed using a Molecular Devices spectrophotometer (Sunnyvale, CA, USA). TL and mtDNA copy numbers were determined using the Absolute Human Telomere Length and Mitochondrial DNA Copy Number Dual Quantification qPCR Assay Kit (Science Cell Research Laboratories, Carlsbad, CA, USA), following manufacturer instructions. The quantitative real time polymerase chain reactions were run using the 7300 CFX Connect Real Time PCR System (Applied Biosystems Biorad, Foster City, CA, USA). Laboratory personnel were blinded to participants’ characteristics, and all assays were processed in triplicate under identical conditions.

### 2.5. Statistical Analyses

The collected data were inserted in an anonymous database. The Shapiro–Wilk test was used to assess normal distribution. The paired *t*-test and Wilcoxon–Mann–Whitney test were used for comparisons of variables obtained before (T0) and after (T12) the intervention depending on the distribution of data. Participants were dichotomized into two groups by their median TL values at baseline: the lower than median group (L-TL group) and the higher than median group (H-TL group). The TL variation (delta) was calculated as the difference between T12 and T0 value; deltas were compared between groups by Wilcoxon–Mann–Whitney test. The correlations between TL variations and clinical and laboratory variables were evaluated using Pearson’s correlation coefficient. GraphPad Prism version 9.0.0 for Windows (GraphPad Software, San Diego, CA, USA, www.graphpad.com, accessed on 26 September 2022) was used. A *p*-value < 0.05 was considered as statistically significant.

## 3. Results

A total of 20 patients were enrolled; 3 of them dropped out for personal reasons. Participants were mostly women (*n* = 13) with class II obesity and central fat distribution. The clinical characteristics are reported in Table 1. At baseline (T0), mean daily caloric intake was 1890.8 ± 398.2 Kcal/die, with a macronutrient distribution of 33.0% lipids, 16.1% proteins, and 49.1% carbohydrates. The median METs values were 29 h/week (interquartile range 14.3–64.5).

The TL values were significantly correlated with HDL-cholesterol (Rho = +0.54, *p* = 0.023), HOMA-IR (Rho = −0.52, *p* = 0.034), and insulin values (Rho = −0.58, *p* = 0.015); the mtDNA copy number was significantly correlated with HDL-cholesterol (Rho = +0.66, *p* = 0.004) and insulin levels (Rho = −0.53, *p* = 0.028) at baseline.

At the end of the study (T12), the mean daily caloric intake was significantly lower (1449 ± 232.1 Kcal/die, *p* < 0.001). Patients lost 6.6 ± 2.9 kg, with a BMI change of −2.62 ± 1.1 Kg/m^2^, a waist circumference reduction of 6.5 ± 6.9 cm, an average fat mass reduction of 1.27 ± 1.68%, and fat-free mass increase of 1.3 ± 1.63%. Hs-CRP circulating levels significantly lowered during the follow-up (*p* = 0.008, Table 1). TL and the mtDNA copy number from baseline (T0) to follow-up (T12) were not significantly different in the whole group (Table 1).

When comparing patients according to their median TL at baseline, no significant difference between H-TL and L-TL groups was evident for baseline characteristics, except for TL values, as expected (Appendix A).

At the end of the intervention (T12), both groups showed statistically significant reductions in weight, BMI and hs-CRP (Table 2). Only in the L-TL group were waist circumference, fat mass percentage, and systolic blood pressure significantly reduced, and fat-free mass and TAC levels increased (Table 2). In the L-TL group, the TL significantly increased at T12, while no significant change was observed in the H-TL group (Table 2).

The variations in weight loss, fat mass decrease, and fat-free mass increase were significantly higher in the L-TL group (Table 3). Furthermore, changes in the TAC levels and mtDNA copy number were significantly higher in the same group than in H-TL patients (Table 3).

## 4. Discussion

In the present pilot study, we observed that after a 1-year multidisciplinary lifestyle intervention, outpatients with obesity and lower than median TL at baseline achieved a greater weight loss, and significantly improved their TL, mtDNA copy number, and TAC levels, when compared to patients with higher than median baseline TL. Furthermore, our data suggested that telomere change is associated with the initial TL.

Indeed, TL has been proven to be a dynamic feature, which can vary in both directions during a lifetime [31]. Aviv et al. [32] in the “Bogalusa study” observed that age-dependent TL shortening was proportional to TL at baseline, underlining the complexity of telomere dynamics in vivo and suggesting that other factors, in addition to the “end-replication problem” and aging “per se”, may influence the telomere length variations. In the Finnish Diabetes Prevention Study, a higher TL increase was observed above all others in pre-diabetic individuals with the shortest TL at baseline after a lifestyle intervention (diet and physical activity) [33]. In patients with severe obesity, an increased telomere lengthening was reported after 6-months intragastric balloon placement in individuals with the shortest TL at baseline [19]. Accordingly, after surgery-induced weight loss, significantly higher telomere lengthening occurred in patients with obesity and the shortest TL at baseline (but not in patients with intermediate or long TL) [31]. Furthermore, baseline TL in adolescents with overweight and/or obesity was an independent predictor of weight loss [18,34,35,36].

The underlying mechanisms responsible for these observations remain to be defined. It could be hypothesized that telomere lengthening is due to the weight loss-induced reduction in oxidative stress and chronic subclinical inflammation. In fact, it is well established that obesity is associated with these two conditions, both accelerating telomere attrition [37]. The L-TL group displayed a significantly higher increase in TAC levels after the intervention, supporting this hypothesis. Telomeres are highly sensitive to hydroxyl radicals, as demonstrated by a review of 22 studies showing that mild oxidative stress accelerated telomere shortening in cultured normal human fibroblasts and endothelial cells, whereas antioxidants and free radical scavengers decreased shortening rates and increased proliferative lifespan [38,39]. Garcia et al. showed a relationship between the dietary total antioxidant capacity and the TL in children and adolescents [35]. Accordingly, reactive oxygen species (ROS) decreased the level of nuclear human telomerase reverse transcriptase (hTERT) and telomerase activity in endothelial cells, leading to the development of a senescent phenotype [39]. Further in vitro and in vivo experiments targeting oxidative stress and telomerases activities in the process of telomere lengthening are necessary to shed light on the underlying implicated mechanisms.

To the best of our knowledge, the present pilot study showed for the first time a significant increase in the mtDNA copy number after a 1-year lifestyle intervention in the L-TL group. The mtDNA copy number is a marker of mitochondrial biogenesis, mitochondrial density, and oxidative capacity [40]. A relationship between mitochondrial-mediated oxidative stress-defense mechanisms and weight change has been suggested in a population-based study, by demonstrating that the leukocyte mtDNA copy number is inversely associated with weight, BMI, waist circumference, and waist–hip ratio [16]. Furthermore, the association between the reduced leukocyte mtDNA copy number, obesity, and metabolic syndrome was widely described in the literature, expanding the spectrum of associations between the mtDNA copy number and metabolic phenotypes in different populations [41]. Telomere lengths and the mtDNA copy number may be additional markers of the dynamic metabolic homeostasis in patients with obesity.

Given the small sample size, these observations are to be considered as a pilot study, and need to be replicated in larger cohort of patients with obesity. At present, TL has mainly been explored in aging and aging effects. Weight variations, abdominal obesity improvement, negative energy balance, and lifestyle modifications could affect the lengthening in a relatively short time.

## 5. Conclusions

The present pilot study suggests that telomere length is extremely variable among patients with obesity. In patients with lower TL at baseline, a 1-year lifestyle intervention led to increased telomere lengthening and weight loss, together with an mtDNA increase and TAC level improvement. Larger longitudinal studies are needed to confirm these results.

## Figures and Tables

**Table 1 nutrients-14-04293-t001:** Characteristics of the study participants (*n* = 17).

	T0	T12	*p*-Value
Age (years)	50.7 ± 11.5	-	
Weight (Kg)	99.7 ± 12.9	93.1 ± 13.5	**<0.001**
Heigt (cm)	158.9 ± 8.8	-	-
Waist (cm)	117.8 ± 8.8	112.4 ± 11.9	**0.007**
BMI (Kg/m^2^)	39.5 ± 3.8	36.8 ± 4.2	**<0.001**
Daily calories (kcal/die)	1890 ± 398.2	1449 ± 232.1	**<0.001**
Fat Mass (%)	45.5 ± 6.1	44.2 ± 6.4	**0.007**
Fat Free Mass (%)	54.5 ± 6.1	55.7 ± 6.4	**0.009**
Fasting Glucose (mg/dL)	88.8 ± 15.7	90.2 ± 8.8	0.680
HbA1c (mmol/mol)	38.8 ± 6.1	37.4 ± 3.9	0.151
HOMA (mmol/L × μU/m)	3.7 ± 2.8	3.7 ± 3.1	0.875
Fasting Insulin (μU/mL)	16.5 ± 11.3	16.2 ± 12.7	0.895
Total Cholesterol (mg/dL)	189.6 ± 28.5	198.7 ± 34.7	0.220
Triglycerides (mg/dL)	114.7 ± 63.4	95.5 ± 42.4	0.067
HDL (mg/dL)	52.1 ± 13.2	57.4 ± 19.5	0.076
hs-CRP (mg/L)	8.0 ± 5.6	5.2 ± 3.8	**0.008**
Diastolic blood pressure (mmHg)	82.6 ± 11.5	84.4 ± 12.1	0.484
Systolic blood pressure (mmHg)	134.4 ± 16.1	125.8 ± 12.7	0.052
MDA (µg/L)	40.1 ± 25.5	51.9 ± 42.1	0.418
TAC (µM/L)	139.0 ± 80.5	166.3 ± 90.1	0.302
mtDNA copy number (*n*)	2439.2 ± 519.8	1266.23 ± 943.9	0.098
Telomeres Length (kpb)	2.32 ± 0.49	1.41 ± 0.30	0.138

Baseline (T0); Twelve months after lifestyle interventionfor weight loss (T12); variables are reported as mean ± standard deviation (SD); *p*-value < 0.05 was considered statistically significant (indicated in bold). Abbreviations: BMI (Body Mass Index); HbA1c (Glycosylated Hemoglobin, Type A1C); HOMA (homeostatic model assessment); hs-CRP (high-sensitivity C-reactive protein);MDA (Malondialdehyde); TAC (total antioxidant capacity); mtDNA (mitochondrial DNA). Data are expressed as mean ± Standard Deviation (SD); a *p*-value < 0.05 was considered statistically significant (indicated in bold).

**Table 2 nutrients-14-04293-t002:** Characteristics of the higher than median group (H-TL, *n* = 9) and lower than median group (L-TL, *n* = 8) before (T0) and after (T12) the lifestyle intervention.

	H-TL (T0)	H-TL (T12)	*p*-Value	L-TL (T0)	L-TL (T12)	*p*-Value
Age (years)	48 ± 11.5	-	-	53.5 ± 11.5	-	-
Weight (Kg)	98.3 ± 13.4	92.3 ± 14.1	**0.001**	101.3 ± 13	94.1 ± 13.7	**<0.001**
Heigt (cm)	158.5 ± 9.1	-	-	159.2 ± 9.1	-	-
Waist (cm)	116.2 ± 9.4	112.3 ± 12.7	0.16	119.5 ± 8.2	113.6 ± 9.6	**0.01**
BMI (Kg/m^2^)	39.1 ± 3.5	36.7 ± 4.4	**0.001**	39.9 ± 3.9	37.1 ± 4.3	**<0.001**
Fat Mass (%)	46.3 ± 5.9	45.1 ± 6.5	0.11	44.6 ± 6.5	43.2 ± 6.5	**0.02**
Fat Free Mass (%)	53.7 ± 5.9	54.9 ± 6.5	0.11	55.4 ± 6.5	56.7 ± 6.5	**0.02**
Fasting Glucose (mg/dL)	85.9 ± 10.7	89.8 ± 10.6	0.18	92.1 ± 7.0	90.8 ± 5.6	0.91
HbA1c (mmol/mol)	37.4 ± 4.5	37.2 ± 3.6	0.80	40.3 ± 7.6	37.7 ± 4.4	0.11
HOMA (mmol/L × μU/m)	2.6 ± 1.2	2.6 ± 1.4	0.91	4.8 ± 1.2	4.9 ± 1.3	0.89
Fasting insulin (μU/mL)	12.3 ± 5.5	12.0 ± 7.1	0.89	21.1 ± 4.4	20.1 ± 6.7	0.97
Total Cholesterol (mg/dL)	187.9 ± 28.5	196.9 ± 40.5	0.27	191.5 ± 30.3	200.8 ± 28.9	0.65
Triglycerides (mg/dL)	96.7 ± 31.1	94.8 ± 30.1	0.46	135.1 ± 37.3	97.7 ± 27.2	0.05
HDL (mg/dL)	57.1 ± 11.9	64 ± 12.5	0.19	46.5 ± 10.8	63.5 ± 12.7	0.06
hs-CRP (mg/L)	9.36 ± 6.5	5.47 ± 3.7	**0.02**	6.58 ± 4.1	4.96 ± 3.5	**0.04**
Diastolic blood pressure (mmHg)	86.1 ± 11.5	86.6 ± 13.5	0.93	83.6 ± 11.5	81.8 ± 10.7	0.35
Systolic blood pressure (mmHg)	136.7 ± 20.5	127.8 ± 12.1	0.23	131.8 ± 9.9	123.7 ± 14.1	**0.01**
MDA (µg/L)	27.6 ± 4.9	60.4 ± 22.1	0.12	54.3 ± 30.2	41.6 ± 8.2	0.37
TAC (µM/L)	151.8 ± 31.7	145.2 ± 38.9	0.19	126.2 ± 36.3	187.2 ± 35.7	**0.01**
mtDNA copy (*n*)	3553.81 ± 780.1	1238.9 ± 438.7	0.07	1185.4 ± 324.2	1296.8 ± 326.8	0.19
Telomeres Length (kbp)	3.47 ± 0.8	1.33 ± 0.8	0.11	1.02 ± 0.4	1.51 ± 0.3	**0.02**

Abbreviations: the lower than median group (L-TL group) and the higher than median group (H-TL group); BMI (Body Mass Index); HbA1c (Glycosylated Hemoglobin, Type A1C); HOMA (homeostatic model assessment); hs-CRP (high-sensitivity C-reactive protein); MDA (Malondialdehyde); TAC (total antioxidant capacity); mtDNA (mitochondrial DNA). Data are expressed as mean ± Standard Deviation (SD); a *p*-value < 0.05 was considered statistically significant (indicated in bold).

**Table 3 nutrients-14-04293-t003:** Changes (deltas) before–after the lifestyle intervention in H-TL (*n* = 9) and L-TL (*n* = 8) groups.

	H-TL (Delta T12–T0)	L-TL (Delta T12–T0)	*p*-Value
Weight (Kg)	−6.1 ± 4.6	−8.1 ± 2.7	**0.03**
Waist (cm)	−3.8 ± 6.8	−6.1 ± 4.9	0.05
BMI (Kg/m^2^)	−2.3 ± 1.7	−3.0 ± 0.9	0.05
Fat Mass (%)	−1.22 ± 1.5	−1.42 ± 1.3	**0.04**
Fat Free Mass (%)	+54.9 ± 5.3	+57.8 ± 6.5	**0.04**
Fasting Glucose (mg/dL)	+3.8 ± 6.5	−1.3 ± 5.5	0.52
HbA1c (mmol/mol)	−0.22 ± 2.24	−2.5 ± 4.3	0.20
HOMA (mmol/L × μU/m)	−0.03 ± 1.11	0.1 ± 1.02	0.52
Fasting insulin (μU/mL)	−0.36 ± 4.9	0.04 ± 3.9	0.72
Total Cholesterol (mg/dL)	+9.0 ± 20.1	+9.4 ± 24.1	0.91
Triglycerides (mg/dL)	−11.9 ± 15.2	−27.3 ± 35.1	0.44
HDL (mg/dL)	+6.8 ± 12.6	+3.5 ± 3.8	0.54
hs-CRP (mg/L)	−3.8 ± 3.4	−1.6 ± 2.8	0.68
Diastolic blood pressure (mmHg)	+0.5 ± 14.3	+1.9 ± 6.5	0.60
Systolic blood pressure (mmHg)	−8.9 ± 12.5	−8.1 ± 7.5	0.50
MDA (µg/L)	+17.2 ± 22.5	−15.2 ± 34.1	0.07
TAC (µM/L)	+36.4 ± 24.1	+58.5 ± 18.6	**0.04**
mtDNA copy (*n*)	−2314.8 ± 724.2	+111.5 ± 478.5	**<0.001**
Telomeres Length (kbp)	−2.15 ± 1.13	+0.57 ± 1.23	**0.04**

Abbreviations: BMI (Body Mass Index); HbA1c (Glycosylated Hemoglobin, Type A1C); HOMA (homeostatic model assessment); hs-CRP (high-sensitivity C-reactive protein); MDA (Malondialdehyde); TAC (total antioxidant capacity); mtDNA (mitochondrial DNA). Data are expressed as mean ± standard deviation, SD; *p*-value < 0.05 was considered statistically significant (indicated in bold).

## Data Availability

Not applicable.

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
