# Peer review of "Telomere Length and Mitochondrial DNA Copy Number Variations in Patients with Obesity: Effect of Diet-Induced Weight Loss—A Pilot Study"

_nutrients, 2022, doi:10.3390/nu14204293_

Round 1

Reviewer 1 Report (Previous Reviewer 2)

Telomere length and mitochondrial DNA copy number variations in patient with obesity: effect of diet-induced weight loss. A pilot study Nutrients,accept

This pilot study  enrolled 20 patients(17 at last) and All patients received dietary recommendations in line with the principles of the Mediterranean diet (MD), with an average calorie intake of ~ 1500 ±100 kcal/die (protein 15-20%; lipids 25-30%; carbohydrates 50-60%, 30g fibers) and the indication for varying foods according to personal eating habits, taste, food preferences, cultural traditions, and work-life habits for 12 months. They found that TL and mtDNA copy number significantly increased in patients with obesity and lower baseline TL values after 1-year lifestyle intervention. Given the small sample size, these observations are to be considered as a pilot study, and need to be replicated in larger cohort of patient with obesity. 

Reviewer 2 Report (Previous Reviewer 1)

good work

This manuscript is a resubmission of an earlier submission. The following is a list of the peer review reports and author responses from that submission.

Round 1

Reviewer 1 Report

The authors have attempted an interesting study.  However, their understanding of the tenets of statistical analysis is inadequate.  Any two data points (such as "before" and "after" means") never reveal a "trend," a term that has been incorrectly applied to the data in line 180.  In addition, because the means are not statistically significantly different, the claims of "increase" or decrease" made in lines 187-189 are not scientifically sound and therefore are speculative and not scientific.

Author Response

Thank you very much for your kind and useful comments. Please find the answers to your suggestions listed below.

The authors have attempted an interesting study.  However, their understanding of the tenets of statistical analysis is inadequate.  Any two data points (such as "before" and "after" means") never reveal a "trend," a term that has been incorrectly applied to the data in line 180.  In addition, because the means are not statistically significantly different, the claims of "increase" or decrease" made in lines 187-189 are not scientifically sound and therefore are speculative and not scientific.

Thank you very much for these kind comments. We are aware that the low number of participants of the present pilot study did not allow to obtain statistically significant before-after differences.e fully agree with your suggestion. The sentence in line 180 “Despite a trend toward a decrease TL and mtDNA copy number variations were not significantly different (Table 1).” is been removed, in according with your suggestion, and substituted by this new sentence “The value of TL and mDNA copy number in total group, recorded at time T12, is not statistically significant vs baseline”.  

Furthermore, to better understand the text, the sentences in lines 187-189 “However the changes in TL were heterogeneous; therefore, the participants were divided in two groups according to TL variations. Seven patients showed a TL increase (iTL group, with a TL increase from 1.76 ± 0.77 at T0 to 2.40 ± 0.17 Kbp at T12), while 10 displayed the opposite (dTL group, from 2.71±0.65 at T0 to 0.72±0.17 Kbp at T12) (Supplementary Figure 1).” have been removed. We introduce this sentence “The changes in TL were heterogeneous. Considering the variation of TL in each patient, we obtained two groups: seven patients (named iTL group) shown a positive variation while in 10 patients no or negative variations were registered (named dTL group). The means ± s.d. were:  iTL group, from 1.76 ± 0.77 at T0 to 2.40 ± 0.17 Kbp at T12 and dTL group, from 2.71±0.65 at T0 to 0.72±0.17 Kbp at T12 (Supplementary Figure 1).” In line with this, we modified also the legend in Supplementary Figure 1: “Telomeres Lenght (TL, kilobase pair, kbp) in each studied patient before (T0) and one year after (T12) the intervention study.  The variations increasing or decreasing trend are indicated by a continuous line for each patient.” 

Reviewer 2 Report

The author choose the patients aged 25-70 years with BMI≥30 kg/m2 and compare the difference  in the improvement in TL and mtDNA  in the studied group before (T0) and after a year (T12) lifestyle intervention (diet + physical activity) in a group of outpatients with obesity. The author found that the significant elongation of TL and increased mtDNA copy number is linked to the weight loss amount and total antioxidant capacity. The topic is very interesting.But the sample size is small and the author didn't design the control group.So this manuscript need revise.

Author Response

Thank you very much for your kind and useful comments. Please find the answers to your suggestions listed below.

The author choose the patients aged 25-70 years with BMI≥30 kg/m2 and compare the difference in the improvement in TL and mtDNA  in the studied group before (T0) and after a year (T12) lifestyle intervention (diet + physical activity) in a group of outpatients with obesity. The author found that the significant elongation of TL and increased mtDNA copy number is linked to the weight loss amount and total antioxidant capacity.

The topic is very interesting. But the sample size is small and the author didn't design the control group. So this manuscript need revise.

Thank you very much for these kind comments. This was a pilot study which is preliminary to a in-progress randomized controlled trial in order to test the effects of different dietary interventions on telomere length.

We are aware that the low number of participants of the present pilot study did not allow to obtain statistically significant differences.

Unfortunately, we are not able to enroll in the due time of revision (10 days) further participants, as the follow-up time is 12-months. All our patients attending the Obesity Unit received lifestyle recommendations, as it is mandatory to try to change these patients' incorrect eating habits. The design of a control group of patients who are left free to continue their usual lifestyle would not have been approved by our ethics committee.

The aim of the present pilot observational study was to assess whether a lifestyle intervention in a real-life situation would be able to modify the telomere length after 1-year follow-up. We fully agree with you that this aim could be unclear in the previous version of the manuscript. We have now re-written the aims in order to clarify the objective of the present study (line 87). Furthermore, the limitations of the study have been better emphasized (line 304). However, we think that this small pilot study, even with its limitations, might add a piece of knowledge to the existing literature on this topic.    

The sentence was changed: “The primary objective of the present pilot study was therefore to evaluate the effects of a lifestyle intervention (diet plus exercise) without specific nutritional or physical activity intervention, but during a real-life observation, on TL and mtDNA copy number in a cohort of adult outpatients with severe obesity.

Round 2

Reviewer 1 Report

Thank you for making the corrections.  Please also remove the misleading phrase "The mean MDA delta values were decreased only in iTL group, despite not significantly." from new lines 204 and 205.

Reviewer 2 Report

I have reviewed this revision of the pilot study.I think the authors have changed the manuscript according the reviewer's comments.I believe the manuscript has been sufficiently improved to warrant publication in Nutrients.